# Autonomous Instrumentation for Measuring Electromagnetic Radiation from Rocks in Mine Conditions—A Functional Analysis

**DOI:** 10.3390/s22020600

**Published:** 2022-01-13

**Authors:** Krzysztof Maniak, Remigiusz Mydlikowski

**Affiliations:** 1National Institute of Telecommunications, ul. Szachowa 1, 04-894 Warsaw, Poland; 2Faculty of Electronics, Photonics and Microsystems, Wroclaw University of Science and Technology, ul. Janiszewskiego 11/17, 50-372 Wrocław, Poland; remigiusz.mydlikowski@pwr.edu.pl

**Keywords:** electromagnetic emission, mine collapse, impulse response, EM receiver, rock destruction, dynamics of operation

## Abstract

This paper analyses the function of an innovative integrated receiver for the measurement of electromagnetic field emissions. The autonomous receiver measures and registers the elevated emission levels of both components of the EM field originating from rocks subjected to increased mechanical stress. The receiver’s sensitivity of 60 µV/m, its dynamic range of 98 dB, and its impulse response of 0.23 V/µs were determined in laboratory conditions. Real EM field signals from hard coal samples subjected to crushing force were recorded using an autonomous receiver. The observed and recorded results confirm that the receiver operates in the full range of amplitudes of the EM field signal emitted from the rock. The results determine the band of characteristic signals for EM field emission from hard coal. The system created on the basis of autonomous EM receivers can support the existing seismic safety systems in real mine conditions by predicting the possibility of mine collapse hazards.

## 1. Introduction

The search for new methods to improve safety at work for miners underground has been going on for years. However, a satisfactory solution to this problem has still not been found. Mine disasters usually result in the loss of miners’ lives or damage to their health. For example, only last year, in 2020, in the mining industry in Poland, which is the leading coal mining country in Europe [1], there were a total of 2029 accidents, of which 16 were fatal. Work stoppages and financial losses resulting from accidents are also not negligible.

Apart from methane or coal dust explosions, the most common causes of disasters are rock bumps, which result in miners being buried by collapsing coal or cutting off a safe retreat route. The effects of underground tremors may also be visible on the surface in the form of shocks felt by people, cracks in buildings, roads, etc. The threat from gases, such as methane, acetylene, and coal dust, in the form of their increased concentrations in the air can be efficiently monitored by using specialized instruments [2,3,4]. As a result, accidents related to gas explosions are reported to be decreasing [5,6,7,8]. Seismic tremors occur when the accumulated energy of generated stresses finds an outlet in the form of rock mass destruction. The standard method used in mines to monitor this type of phenomena is the seismic method [9,10,11,12]. It is based on the recording of seismic vibrations propagated in a coal seam by means of an extensive network of geophones installed at critical points of a mine. The seismic method, due to its poor sensitivity to bumps developing in its initial phase, does not provide full protection for miners. Therefore, new methods are being sought to improve work safety in mining.

It has been discovered that the stressing of rocks and their cracking is accompanied by the emission of pulsed electromagnetic fields. Historical first reports on electromagnetic emission anomalies were related to earthquakes, during which researchers recorded elevated levels of electromagnetic disturbances [13,14,15]. Anomalous signals are recorded over a wide frequency range [16,17]. Research into the phenomenon of electromagnetic emission as a precursor to an impending earthquake is still being developed [18]. An example of such research is reported in [19]. Electric field emission of seismic origin in a 100 m deep borehole was studied. In addition, a second electric field receiver, electrically identical to the underground one, was placed on the ground surface directly next to the measuring well. Digital signal processing made it possible to record weak signals of seismic origin. Moreover, many researchers have reported on anomalously high levels of electromagnetic fields accompanying bouldering processes in hard coal excavations [20,21,22] or increased stresses occurring in the excavation [23]. These phenomena occur even a few hours before a disaster is about to happen, i.e., in advance of the time allowing for efficient evacuation of miners from the place of danger of a collapse [24]. These statements are preceded by laboratory tests of electromagnetic field emissions from rock samples subjected to destructive axial pressure [25,26]. For different types of rock samples tested, frequency bands of increased EM emission can be distinguished, and the time courses of EM field components have the character of fading pulses whose frequency spectrum is continuous [27,28]. 

This article presents the results of work on the system of early warning for the possibility of bumps in the mine on the basis of the measurement and analysis of electromagnetic radiation coming from the coal bed in which mechanical stresses occur. This method, in its current state of advancement, can be an auxiliary method to the seismic method commonly used in mines and fits into the scheme of preventive measures. In the literature, there are reports of studies of electromagnetic field emissions from samples or from hard coal deposits subjected to mechanical loading. An interesting portable device registering the magnetic component of an electromagnetic field, in a very wide band of 1 ÷ 500 kHz, is the KBD 5 device developed by Chinese researchers, which is used for stress measurements in hard coal seams in mines [29]. The angle of measurement of the magnetic field signal of the device is 60°. The prototype of the discussed device, also sensitised to the magnetic component of the field, was the YDD 16 instrument [30]. Another solution used for monitoring the state of stress in walls and vaults of tunnels, including mine workings, is Cereskop manufactured by Ceres GmbH, Staffort Germany [31,32]. The device equipped with a ferrite antenna for the magnetic component enables the recording of signals within the frequency range of 5 ÷ 50 kHz and the maximum amplification of the measuring path at the level of 102 dB.

However, authors of publications on this topic often stop at presenting laboratory experiments without conducting experiments in real mine conditions. Often, the result of electromagnetic field emission from rocks is given in units of voltage read from the output of the measuring system [25], expressed in the number of recorded impulses or energy [26]. Taking the process of microcrack formation as one of the main sources of electromagnetic field emission from laboratory samples, the intensity of deformation processes can be expressed by the accumulated electric charge on the edges of the microcrack [33]. Researchers also often use conventional units that make it impossible to interpret the result at all. It is also common to measure only one component of the electromagnetic field. The ability to independently register the electric and magnetic components of the electromagnetic field can be considered an asset of the receiver presented in this paper. This becomes particularly important in the near field, where there are no unambiguous relations between the components of the electromagnetic field. Taking into account the significant lengths of the electromagnetic waves under consideration, most measurements are made in the near field. However, the greatest advantage of the developed device is the possibility of autonomous, unmanned long-term operation in a mine excavation. This was achieved by developing a control algorithm that keeps the receiver in a sleep state until an electromagnetic disturbance signal is present. Information is stored on a built-in memory card or can be transmitted via an RS232 interface to the outside world for further processing.

A similarly innovative method for the recording of electromagnetic fields is the study of acoustic emission from rocks, including hard coal subjected to strong mechanical stress. As a result, cracks are created whose emission is recorded by sensitive microphones [34,35]. Thus, some similarities between seismic and acoustic methods can be identified, where the main difference lies in the range of frequencies recorded. The testing device counts the acoustic events that occur. The spectrum of the recorded acoustic signals lies below a frequency of 100 kHz [36]. Some reports include results of joint acoustic emission tests and measurements of generated electromagnetic fields from rock samples [34,37]. In many cases, the simultaneous measurement of acoustic emission and electromagnetic radiation can be considered redundant. Acoustic fields are more susceptible to acoustic disturbances from operating mining machinery, for example. The electromagnetic method is more resistant to disturbing electromagnetic fields commonly present in mine workings due to the use of effective electromagnetic field shields. These facts support the superiority of electromagnetic field emission testing over acoustic field emission testing. In addition, acoustic fields are much more strongly attenuated in the rock medium relative to low frequency electromagnetic fields, below 50 kHz. This fact forces denser placement of microphones, making the system more complex. Thus, there is a real indication that the method of investigating anomalous electromagnetic fields in mine conditions will be a valuable complement to studies using traditional mining geological methods.

## 2. Mechanism of Electromagnetic Field Emission from Rocks

The main mechanisms that initiate the generation of electromagnetic field emission under stress within rocks could include:

### 2.1. Piezoelectric Effect

The piezoelectric effect is observed in dielectric media with anisotropic structure, having an appropriate crystallographic structure. Naturally occurring materials that exhibit the piezoelectric effect include: quartz, tourmaline, sphalerite, nepheline. There are also synthetic piezoelectric materials that include: crystals of Seignette salt, potassium hydrogen tartrate, ammonium dihydrogen phosphate (ADD), potassium dihydrogen phosphate (KDP), and barium titanate.

Relationships describing the simple and inverse piezoelectric effect are of the form [38,39]:(1)D=dT+εE
(2)S=sT+dE
where: 

*D*—electric displacement [C/m^2^]

*T*—stress component [N/m^2^]

*E*—electric field component [V/m]

*S*—strain component

*ε*—electrical permeability [F/m]

*D*—piezoelectric constant [C/N]

*S*—elastic stiffness [1/Pa]

The basis for the discussion of electromagnetic field emission from rock samples is the simple piezoelectric effect, described by the relation (1). The piezoelectric phenomenon can occur in materials exhibiting spontaneous or induced polarization. This polarization changes under the influence of mechanical stress T, which causes the electric induction. As a result of the formation of stresses in the rocks in this coal, increased pressure is exerted on the contained piezoelectric material and an electric field is created. The reverse piezoelectric effect described by relation (2) is a phenomenon involving a change in the dimensions of the piezoelectric material under the influence of an applied external electric field.

Many researchers link the piezoelectric effect to developing cracks and stress-vibrations in the rock structure [40,41]. Example relationships describing such a phenomenon are as follows [42]:(3)$$\sigma_{ij}=c_{ijkl}U_{kl}-e_{kij}E_{k}$$
(4)$$D_{i}=\varepsilon_{0}\varepsilon_{ij}E_{j}+e_{ikl}U_{kl}$$
where:

*σ**_ij_*—mechanical stresses tensor

*U_kl_*—displacement vector

*D_i_*—electric displacement [C/m^2^]

*ε*_0_, *ε**_ij_*—electrical permeability [F/m]

*E_j_*, *E_k_*—electric field component [V/m]

*c_ijkl_*—tensor of elastic

*e_ikl_*—piezoelectric tensor

The above equations link the emitted electric field strength E of the emission with the displacement vector U caused by the occurring deformation processes of the rock structure.

### 2.2. Microcracks in the Rock Structure

As a result of increasing mechanical stresses, micro-cracks and fractures are formed in rocks, what is illustrated in Figure 1 [26]. Already existing cracks are also enlarged. The fracture faces exhibit mechanical oscillations. It is worth mentioning that similar phenomena are also observed in plastic materials such as epoxy resin [43].

According to the theory of crack propagation in rock material, the crack width *d* can be determined from the relation [44]:(5)$$d=\frac{\sqrt{K_{1}^{2}+K_{2}^{2}}}{E_{m}}{\left(\frac{l}{2\pi }\right)}^{\frac{1}{2}}\left(1-v^{2}\right)$$
where:

*K*_1_, *K*_2_—stress intensity of the first and second type [Pa]

*E_m_*—flexibility module [Pa]

*L*—crack length [m]

*V*—Poisson’s ratio

As a result of stress, deformation of the rock material and interplanar friction occur, assisted by the movement and transfer of atoms [45]. This leads to the appearance of positive and negative charges at the edges of the fracture.

The equation relating the electric field strength E to the parameters of the micro-crack formed takes the form [46]:(6)$$\Delta E-\nabla divE-\frac{\varepsilon_{ij}}{c^{2}}\frac{\delta^{2}E_{j}}{\delta t^{2}}e_{i}=\frac{1}{\varepsilon_{0}c^{2}}e_{ikl}\frac{\delta^{2}U_{kl}}{\delta t^{2}}e_{i}$$
where: 

*U_kl_*—deformation tensor

*c*—speed of light [m/s]

*ε*_0_—electrical permeability of a vacuum [F/m]

*ε**_ij_*—electrical permeability tensor,

*t*—time [s]

*e_i_*—unit vector.

In the analysis of the phenomenon, we assume that the crack propagates along the *z*-axis. In this situation, the molecular bonds are broken and charges appear on the crack surface. This leads to asymmetric charge separation. Research is still being conducted on the elementary method of the essence of this phenomenon [44]. However, from observations, it can be concluded that the more cracks formed and the faster the crack propagation, the higher the electromagnetic field strength that can be recorded [33].

For a given width *d* of the microcrack, the system can be regarded as an elementary electric dipole formed by a pair of opposite charges, *+q* and *−q*. This situation is illustrated in Figure 2. This causes the formation of an electromagnetic field in a conducting medium, which can be regarded as coal mine seams. Such a medium is characterized by parameters [47]:-magnetic permeability *µ* = *µ*_1_·*µ*_0_ (H/m)-electric permeability *ε* = *ε*_1_·*ε*_0_ (F/m)-electrical conductivity *σ* (S/m)-specific electrical resistance *ρ* = 1/σ (Ωm)

Based on the above parameters, it is possible to introduce an important concept describing the medium in which an electromagnetic wave is propagated, and that is the wave number of the medium [47]:(7)$$\gamma =\sqrt{\frac{i\omega \mu }{\rho }-\varepsilon \mu \omega^{2}}=\sqrt{i\omega \mu \left(\sigma +i\omega \varepsilon \right)}$$

The electromagnetic field components at a distance *r* from the source propagated in a semiconducting medium are described by the relations [48]:(8)$$E_{r}=\frac{-Idz}{4\pi }\cdot 2i\omega \mu cosQ\frac{e^{i\gamma r}}{\gamma^{2}r^{3}}\left(1-i\gamma r\right)$$
(9)$$E_{\theta }=\frac{-Idz}{4\pi }\cdot i\omega \mu sinQ\frac{e^{i\gamma r}}{\gamma^{2}r^{3}}\left(1+\gamma^{2}r^{3}-i\gamma r\right)$$
(10)$$H_{\varphi }=\frac{Idz}{4\pi }\cdot sinQ\frac{e^{i\gamma r}}{r^{2}}\left(1-i\gamma r\right)$$

The near zone, which is the space near the dipole, can be described here. It can be written with a condition |*γ*·*r*| << 1. Similarly, the far zone is defined by the relation |γ·r| >> 1.

In the above equations, the current *I* flowing in front of the electric dipole can also be expressed by the amplitude *q*:(11)$$I=-\frac{d}{dt}\left(qe^{-i\omega t}\right)$$

### 2.3. Turbulent Fluid Flow through Rock Micropores

When rock formations containing pores, including coal deposits, are significantly saturated with water, they can be considered electrically conductive media. Due to the increased stress present in the rock structure, fluid flow occurs in a turbulent manner. Turbulence in the fluid flow can originate from vibrations of the rock material under load. This reflects the change in velocity of water flow depending on the degree of stress in the porous medium. The fluid with instantaneous velocity $\vec{v_{c}}$ flows in the earth geomagnetic field with magnetic induction $\vec{B_{E}}$. Then, a Lorentz force of acts on each electron [49]:(12)$$\vec{F_{L}}=e\cdot \vec{v_{c}}\times \vec{B_{E}}$$
where:

*e* = −1602 × 10^−19^

C—electron charge

*F*—force [N]

$\vec{v_{c}}$—liquid flow rate [m/s]

*B*—magnetic induction [T]

This force contributes to the orderly movement of electrons, inducing an electric field of intensity:(13)$$\vec{E}=\vec{v_{c}}\times \vec{B_{E}}$$

Moreover, the fluid-saturated geological layers forming the coal seam can also be considered at the microscale as a network of capillaries formed in the granular structure of the medium. The fluid filling the capillaries carries ions, which interact with the grains of the medium to form an electrical double layer [50].

Assuming no external current sources and uniformity of the fluid flow through the medium, one can determine the electric potential difference *V*, or voltage, in the circuit in the idle state (*J_C_* = 0) occurring under the influence of pressure changes *p*. This relationship is expressed by the Helmholtz–Smoluchowski relation [50,51,52]:(14)$$\nabla \left(-V\right)=-\frac{L_{21}}{L_{22}}\nabla \left(-p\right)$$
where:

*V*—voltage [V] 

*p*—pressure [Pa]

*L*_21_ = −*ε*ζ/η = ke—electroosmotic coupling factor [m^2^/V⋅s]

*L*_22_ = σ—electrical conductivity of liquid [S/m].

## 3. Laboratory Test Stand for EM Field from Rocks

Laboratory tests of EM field emissions from rock samples subjected to increasing compressive force were conducted on a test rig as shown in Figure 3. The rock sample was subjected to an incremental compressive force through the jaws of a press with a maximum possible compressive force of 450 kN. Different rates of force increment were tested from 100 N/s to 40 kN/s by observing the EM pulses generated from the rock. Increasing the rate resulted in an increase in the number of EM pulses generated associated with the failure of the sample. In each case, the spectrum of the pulses obtained ranged from hundreds of Hz to 30–40 kHz. To ensure proper observation and recording of the phenomenon of the generated EM emission pulses during the development of microcracks and the final destruction of the samples, the optimum crushing force increment was set at 1 kN/s. This rate allows for detailed observation of the developing process of rock destruction without missing important phases of the phenomenon that may occur in too slow or too fast a destruction process.

Separate receivers of the electric component E and the magnetic component H of the EM field were placed in close distance to the tested sample. For isolation from external interference, the measuring receivers together with the tested rock sample were shielded with a screen connected to the grounding system. A steel shield was used to limit the influence of the magnetic component of external disturbances and a parallel copper sheet shield was used to limit the external electric fields. In this way, the composite shield enables independence from external interference over a wide spectrum. The shielding effectiveness increases with increasing interference frequency. The use of the screen reduces the influence of electromagnetic interference by an average of 35 dB in the frequency band up to 100 kHz, which can be considered a satisfactory value [47].

Coal, sandstone, dolomite, and magnesite samples were subjected to destructive testing. Samples from mine and rock boreholes were cut to a height of 5 cm. In order to compensate for unevenness of the specimens and uniform distribution of compressive forces, soft wood was placed between the tested specimen and the jaws of the press. The computer on the measuring stand recorded the amplitudes of the obtained signals of the EM field components [27]. In the further analysis, the spectral distribution of the signals was established. The spectra of the signals for the different types of rocks tested were within the frequency range up to 50 kHz. Significant spectrum bars were observed in the range up to several kHz.

Tests carried out on the test stand provided guidelines for the construction of an autonomous receiver of the arising EM field emission in mine conditions. The receiver should be characterized by:-frequency band: 50 Hz to 50 kHz-minimum sensitivity at E_min._ = 7·10^−5^ V/m and H_min._ = 1.5·10^−5^ A/m, respectively for the electric and magnetic field component receiver-recording the current value and the peak value of the electromagnetic pulse in real time-independent power source for uninterrupted operation of the receiver for a long-term period, e.g., for a month-non-volatile memory-hermetic and spark-eliminating case

## 4. Autonomous EM Field Receiver under Mine Conditions

Figure 4a shows the block diagram of an autonomous EM field receiver. This schematic can be divided into four major functional blocks:EM field electric signal sensor and signal conditioning circuitEM field magnetic signal sensor and signal conditioning circuitdigital signal and control signal processing circuitpower supply system

In order to miniaturize the design, the electromagnetic field receiver was made using SMT technology. The system is located on three printed circuit boards connected with each other by a system of micro-switches. On the base plate, there is a microprocessor with associated circuits and a power supply unit. On the other two boards there are analog signal processing blocks (of the same electrical construction). These blocks cooperate with antennas receiving the electric component, E, and the magnetic component, H, of the electromagnetic field. A view of the receiver’s inside is shown in Figure 4b. After closing the housing, the receiver has the following dimensions: length 500 mm and diameter 40 mm.

The antenna for receiving the electric component was made as a monopole in the form of a 20 cm long copper rod, for which the reference ground is the brass structure of the receiver. At this height of the electric rod antenna, its effective hef is 10 cm, which, if the relation *h_ef_* << *λ*, gives the relation for voltage induced in the antenna [53]:(15)$$SEM_{E}=E\cdot h_{ef}$$
where: *E*—electric field strength [V/m].

A ferrite rod antenna with a coil wound on it was used as the magnetic *H* component antenna. The electromotive force induced in the coil can be determined from the relation [54,55]:(16)$$SEM_{A}=Z_{0}\cdot H\cdot \frac{2\cdot \pi \cdot z\cdot S\cdot \mu }{\lambda }$$
where: 

*H*—magnetic field strength [A/m]

*Z*_0_ = 120π Ω—wave impedance of free space,

*Z*—number of antenna wire turns

*S*—cross section area of antenna coil [m^2^]

*λ*—signal wavelength [m]

*μ*—the magnetic permeability effective value of the antenna’s ferrite core [H/m]

## 5. EM Field Signal Sensor and Signal Conditioning Circuit Block 

Blocks 1 and 2, shown in Figure 4a, are analog processing units for the electromagnetic signal under test. In terms of construction, these blocks have an identical structure for the electric and magnetic components. These systems have been structurally modified in relation to the previously used solution [56]. A simplified schematic diagram of the modified device is shown in Figure 5.

The input signal obtained from the electric or magnetic antenna in the form of *U_INPUT_* voltage goes to the input of the voltage follower *U1A*, whose role is to impedance match the measurement antenna to the other blocks of the device. The next element of the signal processing path is one stage of passive bandpass filtering consisting of *R1*, *C1*, *R2* and *C2* elements. This filter has a frequency response in the range from 84 Hz to 54 kHz. The second filtering element, identical to the first one, consists of *R6*, *C6*, *R7* and *C7* elements. It was decided to build a two-stage (2nd order) filtering system in order to obtain satisfactory steepness of the frequency conversion characteristics. In addition, the use of a passive filter reduces current consumption by the system from the power source.

The transmittance of a bandpass filter section depends on:(17)$$G1=\frac{j\omega R_{2}C_{1}}{1+j\omega \left(R_{1}C_{1}+R_{2}C_{2}+R_{2}C_{1}\right)-\omega^{2}R_{1}R_{2}C_{1}C_{2}}$$

The next elements of the circuit are amplifying levels of the tested signal, realized on operational amplifiers *U2A*, *U3A* and *U4A*. The amplifiers, together with the keys *S1*–*S5* switched by the control processor, function as amplifiers with controlled gain. The transmittance of a single amplifying element, for the key *S1* located as shown in Figure 5, is presented in the relation
(18)$$G2=1+\frac{j\omega R_{5}C_{3}}{1-\omega^{2}R_{3}C_{3}R_{5}C_{5}+j\omega \left(R_{3}C_{3}+R_{5}C_{5}\right)}$$

Controlling the S1–S5 keys provides discrete resultant gains, which are the product of the gains of the individual circuits, with values of: 6, 30, 60, 180, 300, 900, 5400, 9000, and 27,000 V/V. This process is designed to amplify weak signals more and amplify them less in the case of waveforms with large amplitudes. For example, for the highest sensitivity of the electrical component measurement path, which is achieved for a resultant path gain of 27,000 V/V, it can be shown that an antenna output amplitude of 0.1 mV corresponds to an electric field strength of 10 mV/m. The filtered and amplified signal is available at output 1 and is fed through buffer B to the analog input of the processor. In real time, the obtained waveforms of both field components E and H are recorded and stored for a specified time in the dynamic memory of the processor as the instantaneous value of the signals.

By extending the circuit with additional elements and using an appropriate algorithm controlling the operation of the device, it is possible to record anomalous EM waveforms in permanent memory. This is done using the detector built on the *U5A* circuit in cooperation with the Schottky diode, from which the peak value of the waveform is obtained (output 2). The direct signal and its peak value are stored by the processor on the carrier (SD card) under the condition that at the input of the comparator with hysteresis output 3 (*U6A* circuit) appears a voltage with amplitude exceeding the value of the reference voltage *Uref* fixed.

## 6. Determination of Parameters of Analog Signal Processing Block

In order to verify the correct operation of the autonymous receiver in terms of processing the signals measured by the antennas according to the design assumptions, its electrical parameters were determined under laboratory conditions.

### 6.1. Frequency Characteristics of Analog Signal Processing Block

Control measurements were made under laboratory conditions of the real system, which determined the frequency response of the analog receiving circuit of the device. The measurements were performed for different gains (different positions of the *S1*–*S5* keys). Figure 6a shows the amplitude characteristics for the highest gain of the circuit (27,000 V/V) as the least favorable in terms of frequency response. With a gain decrease of 3 dB (i.e., 0.707 of the maximum value), the lower and upper frequency responses of the signals were determined. These are 84 Hz and 54 kHz, respectively. Outside the frequency response, the amplitude characteristic has a slope of 12 dB/oct, which corresponds to the double pole of the system transmittance for the falling edge and double zero for the rising edge. This is the result of a two-stage bandpass filter (2nd filter row). 

Figure 6b shows the phase change of the output signal. The phase changes from +180° to 0° and further to –180°. Such a phase change is characteristic of second order dual pole transmittance bandpass filters [57]. The measurements confirm the design of the 2nd-order filter to ensure selective operation of the system beyond the range of the transmitted signal.

### 6.2. Dynamics of System Operation

To determine the dynamic performance of the autonomous receiver system, the dependence of the output voltage *U_OUT_* as a function of the input voltage *U_IN_* coming from the antenna was measured.

The measurements were conducted in the average frequency range of the system, i.e., the normal operating range of the receiver. The results obtained for f = 5 kHz are summarized in Table 1. The measurements were carried out in the range of linear operation of the receiver.

Based on the data in the table, the transient characteristics were plotted with the marked thresholds of the automatic gain control AGC system (Figure 7). The smallest measurable input signal amplitude of 6 µV was determined from the measurements. The maximum signal that can be recorded is 500 mV. These values allow to determine the dynamics of the system, according to the formula:(19)$$D=20log\left(\frac{U_{INmax}}{U_{INmin}}\right)=98.41\mathrm{dB}$$

The constructed EM field autonomous receiver system allows for the measurement and recording of signals over a large range of changes in the emerging emission.

### 6.3. Impulse Response IR of the System

In laboratory conditions, tests were carried out to determine the speed of operation of the constructed system. The recorded results are shown in Figure 8. During testing of the signal processing circuit, the receiver with a fixed gain of 6 V/V, the signal simulating a single jump with an amplitude of 200 mV was applied to its input (blue color of the waveform). The response of the system to the given excitation was observed on an oscilloscope (red color of the waveform). From the analysis of the response obtained, the rise time of the output waveform was determined (in the range of 10–90%). On this basis, the impulse rezponse IR of the system was determined:(20)$$IR=\frac{\Delta U}{\Delta t}=\frac{0.921V}{4.09\mu s}\approx 0.23\frac{V}{\mu s}$$

Figure 9 shows the signal of the electric component of the EM field recorded by the autonomous receiver, obtained during the laboratory destruction of the hard coal sample. The recorded pulse shows a field strength increase of 4 V/m, which corresponds to a voltage of 400 mV with the assumed E component antenna length. This change took place over a period of 5 ms. The rise rate of the recorded signal is defined as: (21)$$\frac{\Delta U_{IN}}{\Delta t}=\frac{0.4V}{5\mathrm{ms}}=0.08\frac{V}{\mathrm{ms}}$$

During laboratory tests with crushing of rock samples, the recorded waveforms were characterized by a slower rise time than the recording capabilities of the constructed receiver. The response speed of the system is satisfactory from the point of view of recording the waveforms accompanying the emission of the EM field from the rock samples subjected to destruction.

## 7. Tests of Autonomous Receiver in Laboratory Conditions

A stand-alone receiver was placed on the laboratory bench in Figure 3 and a series of tests were conducted with various rock samples. Tests were conducted on sandstone, dolomite, granite, and coal samples. The bench was extended with extended screens to prevent the influence of external EM fields. Representative results for coal samples are shown in this paper.

The tests were conducted on hard coal samples obtained from operating mine workings. The coal samples were obtained from boreholes drilled at the face of the coal walkway. Samples with a base diameter of 2.5 cm and height of 5 cm were placed in the jaws of a hydraulic press. An incremental pressure of 1 kN/s was exerted on the test sample by the jaws of the press. 

Figure 10 shows a representative response of the coal sample in the form of EM emission recorded by the constructed receiver. The time course of the increasing force of the press jaws on the sample is shown (Figure 10a). as well as the time course of the electric component E of the EM field emitted by the coal sample. The real-time waveforms were observed at output 1 (Figure 5) and recorded digitally via the RS 232 interface available in the receiver.

Figure 11 shows the signal of increased EM emission just before the destruction of the tested coal sample at about 38 s of the experiment. The force acting on the coal sample then reaches a value close to 38 kN. The sample was partially destroyed, causing the increased emission of electromagnetic energy in the form of a pulse recorded by the EM field receiver.

The signal reaches inter-peak values in the range of –5–0.5 V/m, which means that the receiver circuit switched between Au = 6 V/V and Au = 27000 V/V gain. Due to the high energy of the recorded EM field, the system activated all gain stages. The tested signal. due to the high level of released energy (anomalous), was stored in the permanent memory of the autonomous receiver and is available on the SD card. The developed receiver control algorithm stored the anomalous signal along with 100 ms of the signal preceding the event.

The determined spectrum for this time interval (Figure 11b) shows dominant signals in the frequency range from 200 to 800 Hz. In this range, the most noticeable frequencies are 450 Hz and 550 Hz.

Figure 12 shows the second recorded pulse of energy accompanying the complete failure of the specimen at the time above 40 s of the experiment. After the initial failure of the tested specimen at the time of about 38 s, a slight decrease of the force acting on the specimen is visible (decrease to about 36 kN). At the time of complete specimen destruction, the highest amount of electromagnetic energy is generated. The generated impulse during the destruction lasts about 300 ms in its initial phase, reaching the value of 5 V/m.

The receiver measuring the resulting energy pulse operated over its full dynamic range, with gains ranging from 6 V/V to 27,000 V/V. In a short pulse, there is also a momentary overdrive of the receiver (pulse energy above 5 V/m). Similar to the waveform in Figure 11, the algorithm used stores the elevated EM emission signal in the receiver’s memory and then saves it to the SD card. The algorithm stored the anomalous signal in memory along with 100 ms of the preceding waveform. 

As in the analyses of previous results, the spectrum of the obtained signal shows dominant frequencies from 200 to 800 Hz with an indication of a distinct 500 Hz signal.

## 8. Conclusions

During verification tests of the receiver’s operation, its signal processing dynamics was determined at the level of 98 dB. The dynamics results from the construction of the receiver (nine discrete amplification values realised on three amplification stages) and the algorithm used to control the receiver. The receiver enables to observe weak signals of the EM field coming from rocks at the level of µV/m, which proves its high sensitivity. It is also possible to record extremely high intensities of EM field emitted by large microcracks formed in close proximity to the receiver.

The system’s response rate to pulse excitation was also determined to be 0.23 V/μs. Taking into account the fact that the rise rate of impulses originating from field emission from destroyed rock samples is at the level of single ms, the response capability of the receiver can be considered fully satisfactory. 

The operation of the autonomous receiver was tested in laboratory conditions on a test bench during the subjecting of a hard coal sample to a destructive crushing force. The EM field emission receiver allowed for the observation and recording of real EM field signals and its gain varied depending on the amplitude of the EM signal tested. The character and frequency bands of the observed signals were analogous to the signals observed in earlier measurements carried out on a laboratory bench by stationary antennas [29]. The receiver meets all the technical assumptions, in particular:-frequency selectivity from 84 Hz to 54 kHz-sensitivity of 60 µV/m-dynamics of signal processing

Laboratory tests have confirmed the formation of electromagnetic emissions from rocks under increased mechanical stress. The spectrum of the resulting signals ranges from hundreds of Hz to single kHz and the intensity to ±5 V/m or ±5 A/m depending on the field component.

The constructed receiver of electromagnetic field components, thanks to its construction, the use of an independent power supply, and the possibility of recording anomalies in permanent memory, can operate in mine conditions and be used to observe rock mass stresses. In the prototype version, the receiver operates independently as a stand-alone device for up to a month. Access to the recorded elevated EM signals is possible in real time via the RS232 interface and after the measurement period by reading information stored on the memory cards.

Further development of the research work carried out in a real mine should allow the construction of a system to support the safety of working miners. An important criterion for the creation of the system is the development of a method for locating similar receivers along the mine workings and linking them into a common monitoring system. In the initial phase of experiments, autonomous receivers should be placed near seismic sensors to study the correlation between seismic and electromagnetic signals.

It is required to know the frequency spectrum of disturbances of operating mining machines in order to eliminate their influence on obtained measurement results. These operations are the easiest to perform using digital filtering algorithms. Additionally, it is necessary to apply appropriate shielding of receivers, which will allow to limit the influence of electromagnetic disturbances coming from the power network and the operating machines.

A system based on independent receivers can support the existing seismic safety systems, forecasting the possibility of mine subsidence

## Figures and Tables

**Figure 1 sensors-22-00600-f001:**
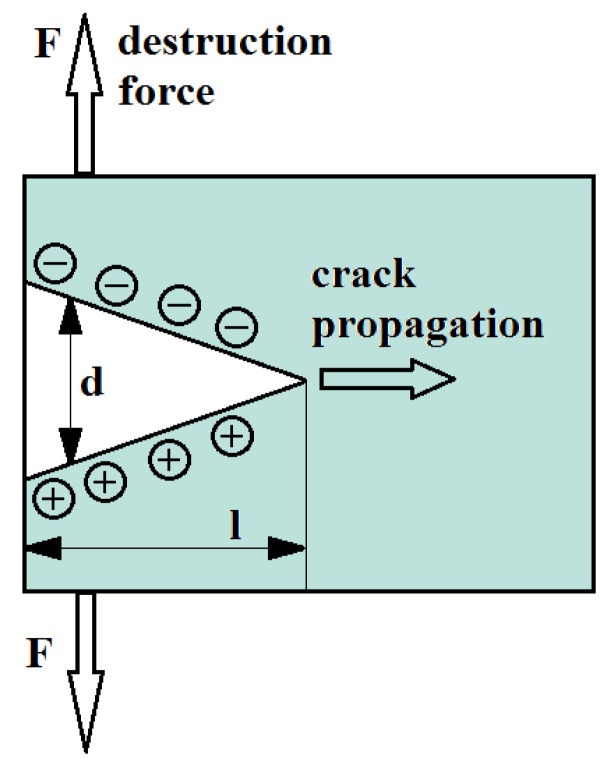
The formation of a crack in the rock structure.

**Figure 2 sensors-22-00600-f002:**
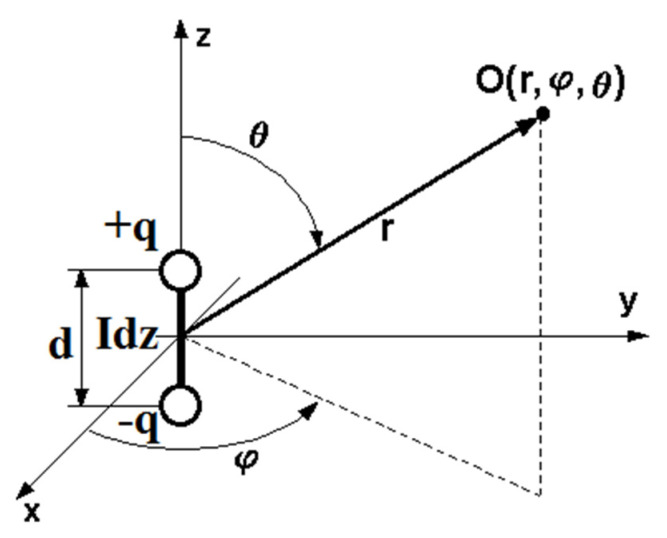
Model of electromagnetic wave propagation from an elementary electric dipole.

**Figure 3 sensors-22-00600-f003:**
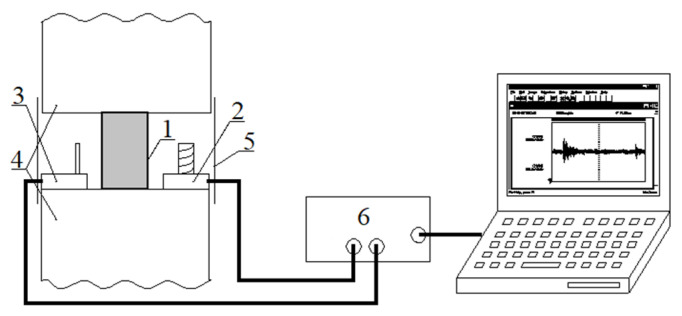
Measuring system for recording electromagnetic fields from rock samples subjected to destructive loading 1—test sample, 2—magnetic field receiver, 3—electric field receiver, 4—hydraulic press, 5—electromagnetic screen, 6—measuring card.

**Figure 4 sensors-22-00600-f004:**
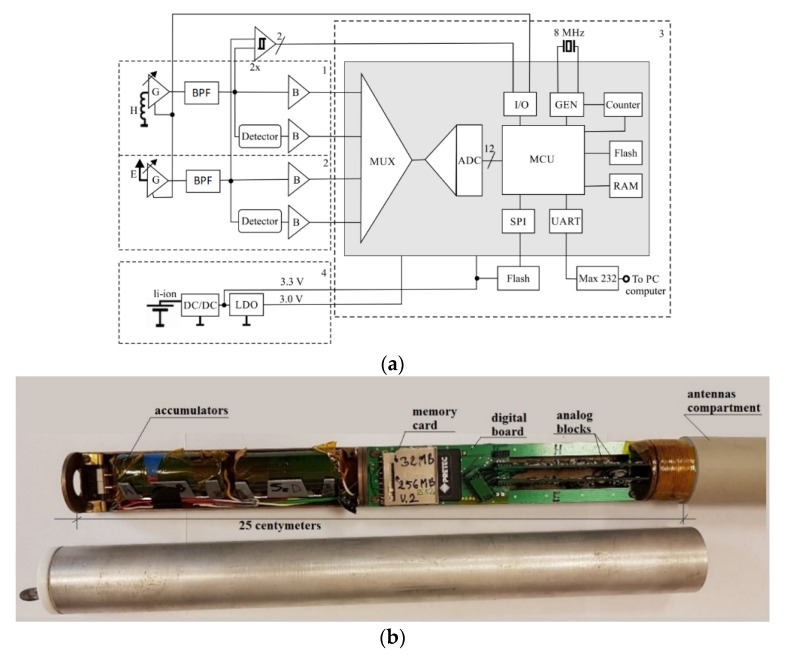
Autonomous EM field receiver. (**a**) block diagram of an integrated EM receiver, (**b**) inside the receiver unit.

**Figure 5 sensors-22-00600-f005:**
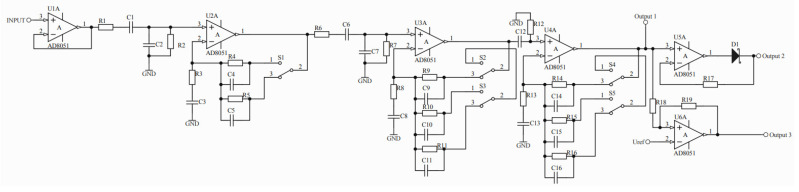
Analog signal processing block diagram.

**Figure 6 sensors-22-00600-f006:**
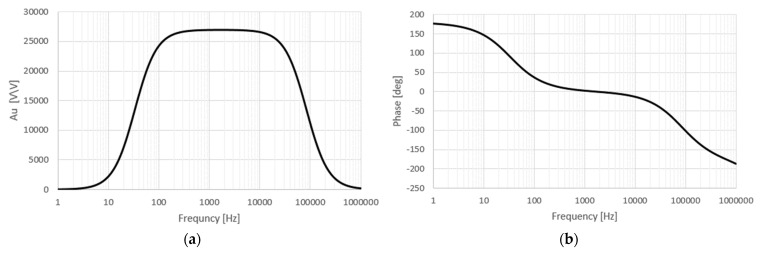
Frequency characteristics of the electromagnetic field receiver: (**a**) amplitude for Au = 27,000 V/V, (**b**) phase.

**Figure 7 sensors-22-00600-f007:**
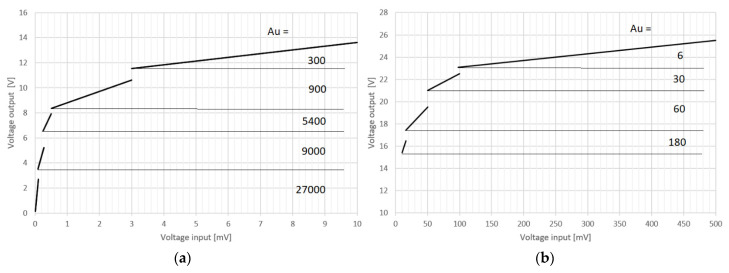
Output voltage as a function of system input voltage. (**a**) system dynamics for gains of 300 to 27,000; (**b**) system dynamics for gains of 6 to 180 V/V.

**Figure 8 sensors-22-00600-f008:**
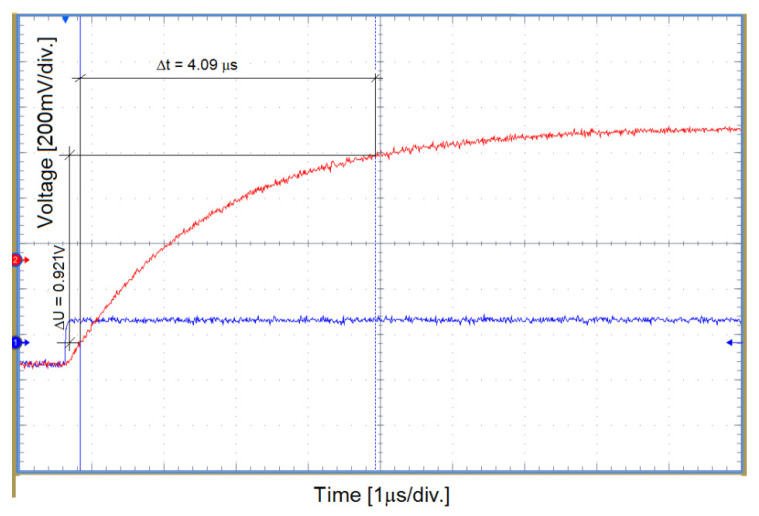
Determination of impulse response IR of the receiver circuit.

**Figure 9 sensors-22-00600-f009:**
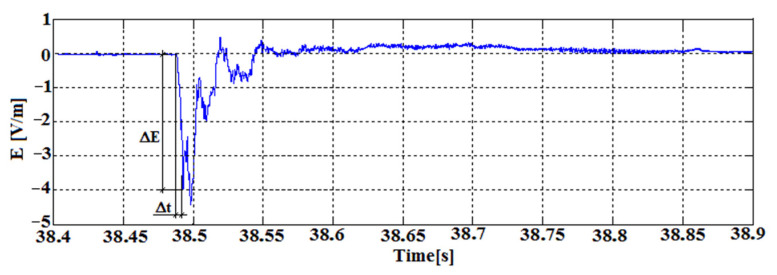
Example of electromagnetic emission signal recorded on the laboratory bench by the autonomous receiver.

**Figure 10 sensors-22-00600-f010:**
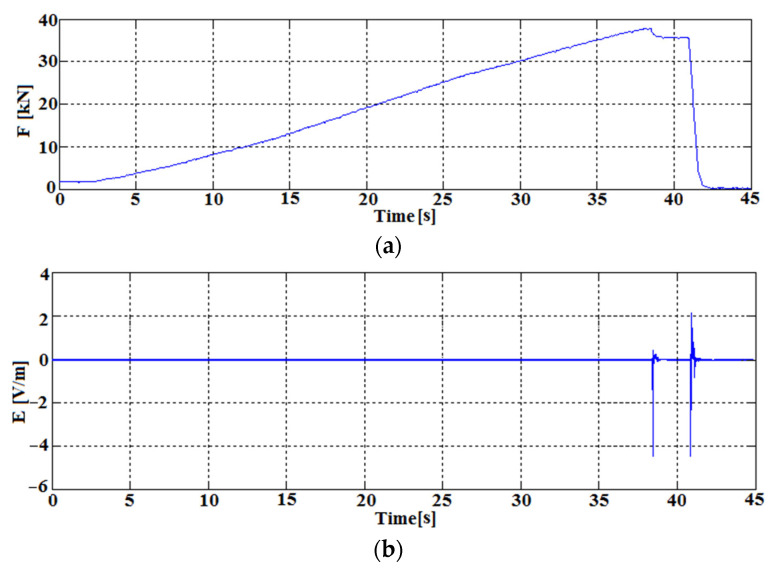
Electromagnetic emission tests on a laboratory test bench. (**a**) time course of the compressive force on the hard coal sample; (**b**) time course of the electric component of the electromagnetic field.

**Figure 11 sensors-22-00600-f011:**
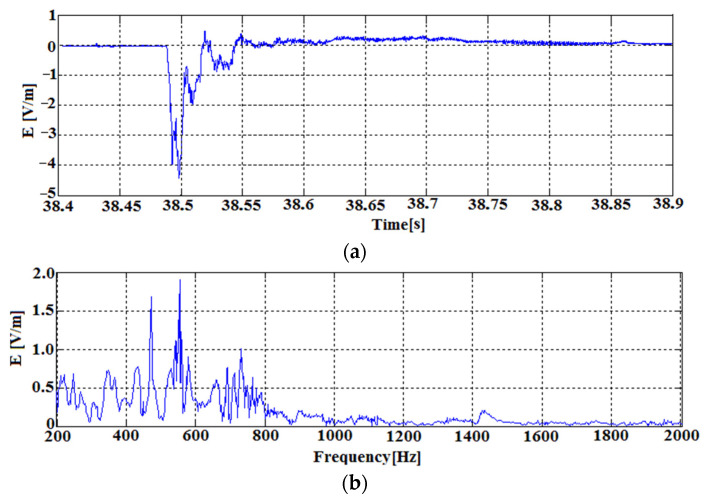
The first of the recorded pulses from Figure 10 at 38.4 to 38.9 s. (**a**) time course; (**b**) spectrum of the waveform.

**Figure 12 sensors-22-00600-f012:**
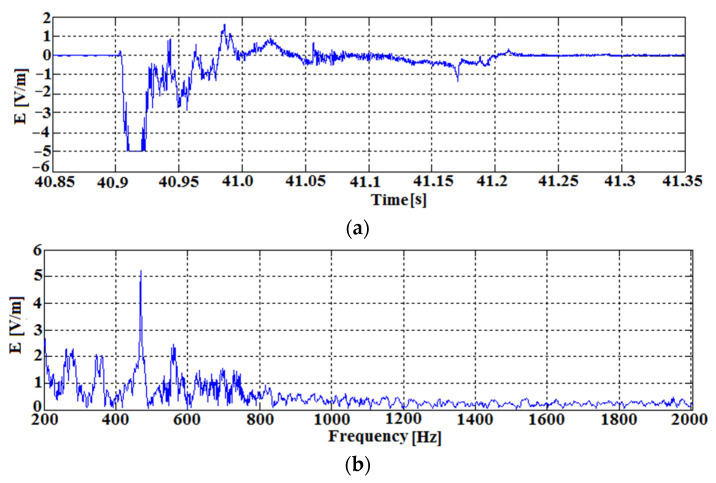
The second recorded pulse from Figure 10 at 40.85 to 41.35 s. (complete destruction of the sample): (**a**) time course; (**b**) spectrum of the course.

**Table 1 sensors-22-00600-t001:** Results of measurements of dynamic.

No.	*U_IN_* [mV]	*Au* [V/V]	*U_OUT_* [V]	*E* [V/m]	No.	*U_IN_* [mV]	*Au* [V/V]	*U_OUT_* [V]	*E* [V/m]
1	500	6	3.000	5.00	10	3	300	0.900	0.03
2	98	6	0.588	0.98	11	3	900	2.700	0.03
3	100	30	3.000	1.00	12	0.51	900	0.459	0.051
4	50	30	1.500	0.50	13	0.5	5400	2.700	0.050
5	50	60	3.000	0.50	14	0.25	5400	1.350	0.025
6	16	60	0.960	0.16	15	0.28	9000	2.520	0.028
7	16	180	2.880	0.16	16	0.09	9000	0.810	0.009
8	10	180	1.800	0.10	17	0.1	27,000	2.700	0.010
9	10	300	3.000	0.10	18	0.006	27,000	0.162	0.00006

## Data Availability

The data obtained during the research are stored on computers belonging to the National Institute of Telecommunications and the Wroclaw University of Science and Technology without external access to the Internet.

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
