# Peer review of "Autonomous Instrumentation for Measuring Electromagnetic Radiation from Rocks in Mine Conditions—A Functional Analysis"

_sensors, 2022, doi:10.3390/s22020600_

Round 1

Reviewer 1 Report

Please refer to the uploaded document

Author Response

At the outset, the authors would like to thank the reviewer for a thorough analysis of the article and valuable conclusions. As suggested by the reviewer, the volume of the introduction has been reduced to include relevant information from the perspective of the subject of the article. The introduction also includes a more detailed discussion of the acoustic method used to detect microcracks in rock structures. Its limitations and lesser usefulness in relation to the proposed electromagnetic (EM) field recording method are presented. At this point, the authors would like to point out that it was not their aim to prove that the proposed method of EM emission testing can replace the methods of excavation stability testing used so far. The proposed method at the present stage of development can be an auxiliary method to the commonly used seismic method. In the introduction, more important commercial excavation stability testing methods are presented.

The authors agree that the presented mechanism of electric field generation explained by the piezoelectric phenomenon has been superficially presented. The content of the article has been supplemented with relevant correlations and, as the reviewer wanted, with literature citations.

The authors' intention was that the content of the article should focus on the functional construction of the autonomous receiver for testing the emitted EM fields from rocks and especially from coal samples. The literature lacks a detailed presentation of the construction and measurement problems that should be faced by a receiver adapted for this purpose. For this reason, the article can be regarded as innovative. Precise values of the intensities of the emitted EM fields are given, which is a frequent lack in the articles in this field.

In the part of the article dealing with laboratory tests it was pointed out that the presented results are representative for the whole series of similar measurements made on hard coal samples. In accordance with the reviewer's recommendations, the method of construction of the electromagnetic shield is given, specifying its attenuation of disturbances in the low frequency range. The screen of similar construction should be used in measurements in a mine.

All figures have been corrected according to the reviewer's recommendation.

The conclusions are extended by the presentation of measurement problems that can be encountered when using the receiver in mine workings. These include mainly EM disturbances originating from mine machinery and equipment. Ultimately, it is planned to build an early warning system based on the presented receiver.

The authors hope that the answer is satisfactory to the reviewer.

Reviewer 2 Report

This paper investigates that autonomous instrumentation for measuring electromagnetic  radiation from rocks in mine conditions. The paper thinks interesting, but the paper needs some modifications.

  1. The author should explain how did the electromagnetic radiation produce form rocks?
  2. The experimental data detected from  autonomous instrumentation is lack, please present more detailed data.
  3. The conclusion should describe more detailed.
  4. In section 2, why did the "Microcracks in the rock structure" and "Turbulent fluid flow through rock micropores" could be detected the radiation.

Author Response

At the outset, the authors would like to thank the reviewer for his thorough analysis of the article and his valuable conclusions.

The reviewer's main concern is the inadequate presentation of how emissions form in rocks under increased mechanical stress. The authors gave an answer to this question in section 2 of the article. Three basic mechanisms of electromagnetic (EM) field generation are indicated. Under the influence of increasing stress in the rock medium, piezoelectric phenomena and rock fractures forming cracks with vibrating walls make themselves known. These processes result in the occurrence of an electrical potential difference, which causes the emission of an electric field. In addition, the potential difference causes a current to flow through the resistance of the medium, resulting in a magnetic field. Similar phenomena are induced by the flow of fluids, especially water through micropores, which vibrate mechanically as stresses develop in the rock structure. These vibrations cause turbulent fluid flow through the pores and the emission of alternating electric voltages which are then transformed into an emitted alternating electric field.

The conclusions are extended by the presentation of measurement problems that can be encountered when using the receiver in mine workings. These include mainly EM disturbances originating from mine machinery and equipment. Ultimately, it is planned to build an early warning system based on the presented receiver.

The authors hope that the answer is satisfactory to the reviewer.

Round 2

Reviewer 2 Report

All comments have modified and replied. The paper could be accepted as this revised form.